# AgentFold: Self-Evolving Exploration of Protein Folding Models

## Abstract

Protein folding, which involves predicting a protein's three-dimensional structure from its amino acid sequence, has advanced rapidly, yet improving modern folding systems remains difficult because pipelines are tightly coupled, evaluation is multi-objective, and progress often depends on coordinated, code-level architectural edits across a large combinatorial design space. We present **AgentFold**, a multi-agent, LLM-driven framework that autonomously proposes, implements, and evaluates modifications to protein folding models, accumulating experimental evidence in a structured memory and using tree-search to allocate compute across model variants. On an engineering-scale folding codebase ($>$ **2,000** LOC), where the implementation complexity induces a vast space of plausible architectural interventions, we conduct a large-scale closed-loop search spanning $\sim$ **80** variants ($\sim$ **5,000** GPU-hours; **170** million LLM tokens), and AgentFold discovers multiple variants that deliver consistent, incremental gains over a strong baseline. More importantly, aggregating outcomes across successful and failed edits exposes implicit design principles of stable folding architectures: improvements repeatedly come from **soft, learnable priors injected early (e.g., sequence/pair biases)** and **gated refinement that regularizes iterative coordinate updates**, while **hard geometric perturbations and geometry-dependent feedback frequently destabilize training**. These results indicate that autonomous, closed-loop optimization can serve not only as an automation tool, but also as a systematic probe that reveals actionable architectural priors for protein structure prediction.

## 1. Introduction

AI scientists aim to automate substantial portions of the scientific workflow by combining large language models (LLMs) with agentic systems (Lu et al., 2024). Recent progress has demonstrated promise across literature review (Hsu et al., 2024), hypothesis generation (Qi et al., 2023), and automated experimentation in code (Tian et al.; Huang et al., 2024). In biomedicine, existing systems largely fall into several categories: (i) *reasoning-centric* agents such as BioReason (Fallahpour et al., 2025) and POPPER (Huang et al., 2025a) that interpret biological evidence and test hypotheses, but typically do not output executable model-development pipelines; (ii) *wet-lab planning* agents such as PerturboAgent (Hao et al., 2025) that optimize experimental decisions (e.g., sequential Perturb-seq design) rather than improving computational models; and (iii) *workflow orchestration* agents such as Biomni (Huang et al., 2025b) and SpatialAgent (Wang et al., 2025a) that connect existing software tools, but are often constrained by predefined toolsets and limited code-generation capability. STELLA (Jin et al.) moves beyond static orchestration via autonomous tool discovery and template learning, yet its scope is primarily lightweight tool use and biomedical question answering, rather than *designing and improving* complex AI models through closed-loop *in silico* experimentation. These gaps motivate a central question: can an autonomous agent go beyond using tools to *iteratively improve* a large, coupled biomedical ML system by proposing, implementing, and validating code-level changes, while also *serving as a systematic probe that reveals the design discipline underlying robust architectures*?

We study this question in the context of *protein folding*, which aims to predict a protein's three-dimensional (3D) structure from its one-dimensional amino acid sequence. Accurate structure prediction enables downstream applications in functional annotation (Todd et al., 2001; Chandonia & Brenner, 2006), protein engineering (Huang et al., 2016; Anishchenko et al., 2021), and drug discovery (Santos et al., 2017). Although modern deep learning approaches have dramatically advanced folding accuracy (Jumper et al., 2021; Makhatadze, 2021), further progress is difficult in practice: constraints are complex, evaluation is inherently multi-objective (e.g., TM-score and lDDT), and state-of-the-art folding pipelines couple many components. Consequently,

---

[1]Anonymous Institution, Anonymous City, Anonymous Region, Anonymous Country. Correspondence to: Anonymous Author <anon.email@domain.com>.

Preliminary work. Under review by the International Conference on Machine Learning (ICML). Do not distribute.

improving a folding system is not merely hyperparameter tuning; it typically requires coordinated, code-level modifications across multiple interacting modules, followed by expensive training and careful interpretation using domain knowledge from both ML and structural biology. This development process remains largely human-driven and hard to scale, leaving much of the model-design space underexplored.

In this work, we introduce **AgentFold**, a fully autonomous multi-agent system for *code-level, closed-loop optimization* of deep learning–based protein folding models that also *extracts reusable design discipline* from intervention–outcome traces. Starting from ESMFold (Lin et al., 2022), AgentFold iterates over a propose–implement–evaluate loop: it scores variants with standard metrics (e.g., TM-score, lDDT), retrieves relevant evidence from a curated folding-model zoo and a structured knowledge base, forms hypotheses grounded in prior outcomes, and generates executable code implementing architectural or algorithmic edits. All runs, including failures, are logged with configurations, diffs, and analyses, which enables cross-variant comparisons that distill recurring design rules.

To support long-horizon improvement under limited compute, we cast optimization as an *MCTS-style search* over concrete model implementations. Each node is a runnable code snapshot, and each expansion instantiates five modules, which are *selection*, *evolution*, *experimentation*, *analysis*, and *storage*, and which together balance exploration and exploitation. On an engineering-scale codebase ($> 2,000$ LOC), AgentFold discovers multiple novel variants with consistent (though modest) gains over the ESMFold baseline. Beyond improvements, aggregating evidence across the search tree reveals which intervention classes tend to yield stable progress (e.g., soft, learnable priors and gated refinement) and which frequently destabilize training, positioning autonomous optimization as both an engineering tool and a systematic probe of folding-model design.

**Contributions.** Our main contributions are:

- We propose **AgentFold**, a fully autonomous multi-agent system for code-level, closed-loop optimization of protein folding models starting from ESMFold, while distilling reusable design discipline from intervention–outcome traces.

- We introduce a self-improving workflow that couples a curated folding-model zoo with a structured knowledge base to accumulate, reuse, and generalize evidence across iterations, including both successes and failures.

- We cast iterative improvement as an MCTS-style search over concrete model implementations, which is instantiated by five modules that handle selection, evolution, experimentation, analysis, and storage, and which enables compute-aware exploration and systematic comparison of competing design choices.

- We demonstrate end-to-end feasibility on an engineering-scale folding codebase ($> 2,000$ LOC), where AgentFold repeatedly finds novel variants with consistent gains on TM-score and lDDT and exposes recurring patterns of stable versus destabilizing edits.

## 2. Related Work

### 2.1. Autonomous AI Research

Autonomous AI Research aims to automate and accelerate the research workflow, from assistant-like tools to increasingly autonomous "AI scientists" (Lu et al., 2024). Early work focused on copilots for coding and experimentation, and later efforts moved toward autonomous hypothesis generation and scientific ideation (Boiko et al., 2023; Tshitoyan et al., 2019). Recent systems further close the loop by iteratively proposing, executing, and selecting improvements, including LLM-guided program evolution (Cheng et al., 2025; Novikov et al., 2025) and autonomous theorem proving with strong verification signals (Chervonyi et al., 2025; Trinh et al., 2024) and reinforcement learning (Oh et al., 2025). In parallel, self-referential paradigms study agents that modify and validate their own code (Schmidhuber, 2006; Baum, 2004; Zhang et al., 2025). Our work is most closely related to ASI-ARCH (Liu et al., 2025), which applies self-evolution to neural architecture design under expensive evaluations.

In contrast to prior self-evolving optimization primarily studied on standard ML benchmarks, AgentFold targets protein folding, where the codebase is large and tightly coupled, constraints are domain-specific, and evaluation is multi-objective. Beyond selecting strong variants, AgentFold also distills reusable design principles by linking code interventions to their observed outcomes.

### 2.2. Protein Folding

Following the breakthroughs of AlphaFold2 (Jumper et al., 2021) and RoseTTAFold (Makhatadze, 2021), a substantial line of work has continued to advance learning-based protein structure prediction (Li et al., 2022; Wu et al., 2022; Cheng et al., 2022; Lin et al., 2022; Ahdritz et al., 2023; Baek et al., 2023; Wohlwend et al., 2025). AlphaFold2 introduced domain-specific architectural components, most notably triangle updates and triangle attention, and it explicitly coupled single and pair representations while leveraging MSAs to inject evolutionary information. Subsequent approaches such as OmegaFold (Wu et al., 2022) and ESMFold (Lin et al., 2022) reduce or remove dependence on MSAs by using embeddings from pretrained protein language models, improv-

ing inference efficiency and extending coverage to proteins with few homologs. In parallel, systems-oriented efforts focus on scaling and efficiency, e.g., FastFold (Cheng et al., 2022) and MiniFold (Wohlwend et al., 2025), which accelerate training and inference through optimized implementations of AlphaFold-style modules. Beyond architectural and systems improvements, recent methods also revisit the generative objective for folding, shifting from deterministic regression to flow-matching/diffusion-style formulations, including AlphaFlow/ESMFlow (Jing et al., 2024) and the flow-based Transformer SimpleFold (Wang et al., 2025b).

## 3. Method

We view autonomous folding-model development as a *search over code-level interventions* and their measured outcomes. AgentFold is designed to produce two coupled artifacts: (*i*) improved model variants and (*ii*) a growing body of *design discipline* distilled from intervention–outcome traces. We cast iterative improvement as MCTS on an evolving variant tree, enabling compute-efficient exploration and controlled comparisons among competing design choices. A self-evolving multi-agent loop proposes and implements edits, evaluates variants, and recovers from failures. Finally, an attribution-and-retrieval stage writes structured intervention artifacts to a database-backed memory, while periodic re-scoring updates node values and refines the search policy. Prompts and templates are in Appendix C.

### 3.1. Problem Formulation & Overview

Given a base folding model $\mathcal{M}_0$ (ESMFold (Lin et al., 2022)), we aim to discover variants $\{\mathcal{M}_t\}_{t=1}^T$ that improve target evaluation metrics and, in parallel, to induce a reusable set of architectural principles $\Pi$ from repeated intervention evidence. Each iteration logs a structured *intervention trace* that records the parent variant, the typed edit (e.g., priors, refinement control, geometry operations), the code diff, stability signals, and metric deltas, which supports cross-variant attribution and principle mining.

To address the complexity of the ESMFold codebase, we propose AgentFold, an LLM-based multi-agent framework grounded in Monte Carlo Tree Search. As illustrated in Figure 1, AgentFold operates via a dual-loop mechanism:

- **Inner Exploration Loop:** A continuous cycle of Sampling, Evolution, Experiment, and Analysis that rapidly generates and verifies new model variants.

- **Outer Periodic Update:** A "Periodic Update" mechanism (e.g., every 10 iterations) that refines the search tree and candidate sets using a composite Scoring Function.

A central Database & Meta Data module serves as an *intervention memory*: it stores runnable code snapshots, code diffs, configurations, logs, and structured attributions, linking them to retrieved literature so that future edits can be proposed and evaluated in a principle-aware manner.

### 3.2. MCTS-based Dynamic Sampling

The search process begins with the Experience Pool (Search Tree), which structurally organizes model variants.

**Top-k Sampling Strategy.** Importantly, sampling multiple siblings from the same parent node creates near-controlled comparisons (holding most code constant), which makes it easier to attribute gains/losses to specific intervention types and to consolidate recurring design discipline. At the start of each inner loop, the Sampler selects nodes from the tree based on a Top-k strategy. This balances exploitation (selecting high-performing nodes) and exploration (selecting diverse reference nodes).

**Context Summarization.** The Summarizer prioritizes evidence that is comparable to the current parent node (e.g., similar edit types or failure modes), producing a compact brief that highlights what previously worked, what failed, and what principles are currently supported by the accumulated traces.

### 3.3. Self-Evolving Agentic Workflow

The Evolution phase transforms the summarized context into executable code through a specialized agent chain:

1. **Deduplication.** First, a Deduplicator Agent screens the proposed optimization direction against historical data to prevent redundant experiments.

2. **Unified Planning & Coding.** Valid proposals are passed to the Unified Planner. Unlike decoupled approaches, this agent is solely responsible for both architectural design and code implementation, ensuring context coherency.

3. **Interactive Debugging.** The generated code enters the Training Environment. A Debugger agent monitors the process in real-time. Upon detecting an Error/Log, the Debugger autonomously interacts with the Unified Planner to iteratively fix syntax or runtime errors until the training launches successfully.

### 3.4. Attribution, Principle Mining & Knowledge Retrieval

Once training concludes (or definitively fails), the system initiates a two-stage post-processing phase to enrich the **Database & Meta Data:**

1. **Automated analysis.** The Trainer streams logs to an Analyst agent, which diagnoses the causal drivers of met-

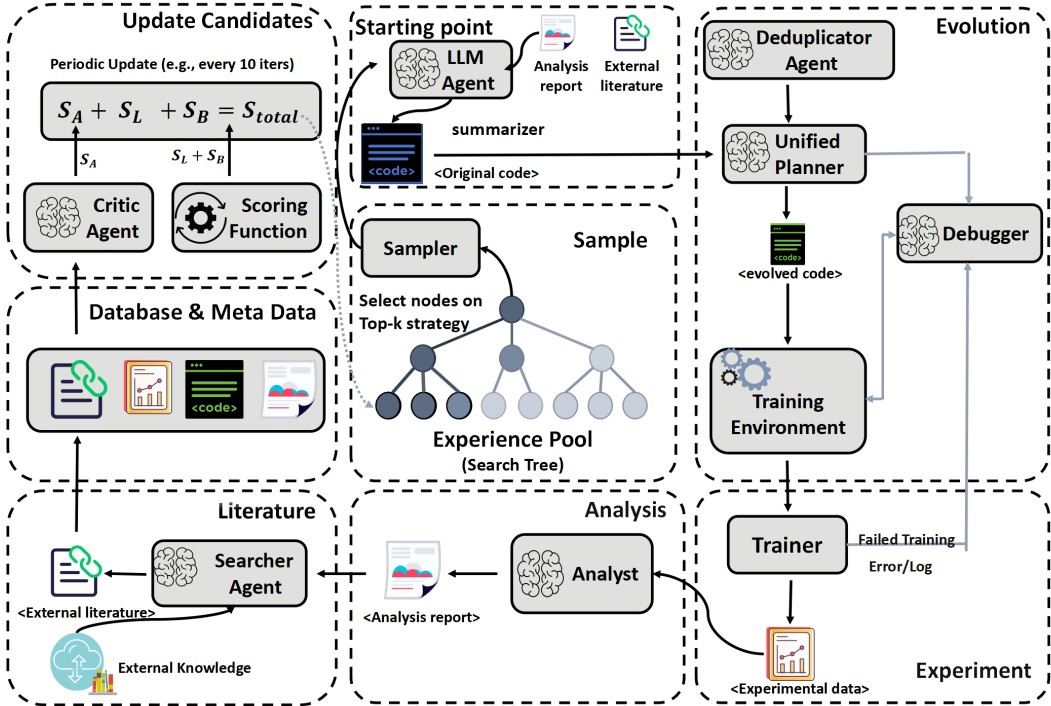

Figure 1: AgentFold system overview. We cast model improvement as MCTS over a code-variant tree, coupling an inner loop (sample → evolve → run → analyze) with a database-backed memory, and an outer periodic update that re-scores candidates to refine the search policy.

ric/stability changes and produces a structured report: attribution of deltas to the intervention and an evidence-based update to the current candidate principles (support-/refute/qualify). Reports are persisted to the database.

2. **Literature augmentation.** In parallel, a Searcher agent monitors new records, retrieves relevant external literature, and links it to the corresponding interventions and observed failure modes. This continuously grounds emerging principles in prior work and provides mechanistic context that transfers beyond a single run.

### 3.5. Periodic Update & Scoring Mechanism

Unlike traditional MCTS that updates the tree immediately, AgentFold employs a Periodic Update strategy (e.g., every 10 iterations) to stabilize the search direction. Scoring Function. We employ a hybrid evaluation module depicted as the "Update Candidates" block. A Scoring Function (algorithmic metric calculator) and a Critic Agent collaboratively compute the total score $S_{total}(e)$:

$$S_{\text{total}}(e) = S_L(e) + S_B(e) + S_A(e)$$

• **Objective Metrics ($S_L + S_B$):** The Scoring Function automatically extracts the Loss Score ($S_L$) and Benchmark Score ($S_B$) from the training logs stored in the database.

• **Subjective Metric ($S_A$):** The Critic Agent reviews the intervention rationale and implementation risk (e.g., coherence with prior evidence, clarity of hypothesis, and likelihood of destabilizing training), yielding an agent score $S_A$ used only to prioritize expensive experiments rather than to claim final improvements..

**Tree Refinement.** At the end of each period, these scores are aggregated to update the node values in the Experience Pool. This periodic synchronization allows the global search policy (Top-k strategy) to evolve based on a batched, robust assessment of recent explorations.

## 4. Main Results

### 4.1. Experiment Setup

**Baseline construction** Our starting point is ESMFold (Lin et al., 2022). We adopt it as the baseline because it is a strong, widely used single-sequence structure predictor that does not require MSA features, making both training and inference substantially more efficient than MSA-based baselines. Moreover, ESMFold retains the core design patterns of modern protein folding systems, which include an Evoformer-style trunk that iteratively refines features and a Structure Module that generates coordinates, and this makes

it a representative and stable reference for controlled comparisons (see Appendix A.1 for more details).

**Mini-data curation** The full dataset consists of protein chains from PDB (Berman et al., 2000; wwp, 2019; Armstrong et al., 2020). Following ESMFold (Lin et al., 2022), we apply a temporal split at 2020–05–01, yielding 518,495 training chains. To obtain a lightweight yet diverse subset, we construct a 1,000-chain mini-dataset via two steps: (i) filter chains released before the cutoff date, and (ii) perform weighted sampling that upweights underrepresented structural clusters using MMseqs2 (Steinegger & Söding, 2017) and prefers medium-length sequences (256–512 residues). This procedure produces a computationally efficient subset while retaining much of the structural diversity of the full training set. see Appendix A.2 for the details.

**Evaluation setup.** We evaluate all variants on CAMEO2022 (Nadeau) following the protocol of (Jing et al., 2023). CAMEO2022 contains 183 proteins with lengths ranging from 100 to 750 residues. For model selection and reporting, we track backbone lDDT (bb_lddt), lDDT (lddt), oligomeric GDT-TS (oligo_gdtts), RMSD (rmsd), and TM-score (TM_score). All metrics are computed with OpenStructure (Biasini et al., 2013) (see Appendix A.3 for the detailed metric definitions). To provide a holistic assessment of model performance, we employ the **Normalized Weighted Relative Score (NWRS)**, a composite scalar that aggregates multiple structural metrics relative to a fixed baseline (see Appendix A.4 for the details).

## 4.2. Main Results

### 4.2.1. QUANTITATIVE ANALYSIS OF MONTE CARLO TREE EVOLUTION

Figure 2 visualizes an MCTS trajectory where each node is scored by the average lDDT (lddt_mean). Darker shading corresponds to higher-scoring variants, whereas gray marks low-scoring candidates with lddt_mean < 0.1 (which are not necessarily collapsed). The tree reveals a pronounced selection pattern: early rollouts sample a wide variety of edits with heterogeneous outcomes, while subsequent search progressively reallocates computation toward branches that consistently produce better structural quality. As a result, low-performing regions are rapidly deprioritized, and the search concentrates on a smaller set of promising neighborhoods. Overall, the observed shift from broad exploration to focused exploitation qualitatively suggests that MCTS acts as an effective filter, steering rollouts away from unproductive edits and toward stable modification trajectories that repeatedly yield improved lDDT.

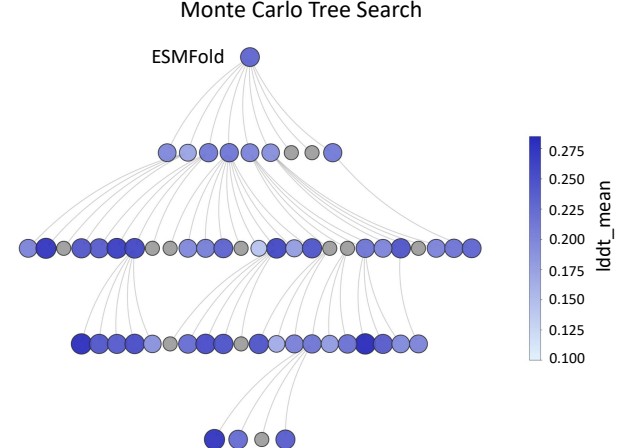

Figure 2: **Monte Carlo tree evolution.** Each node is a sampled variant scored by average lDDT (lddt_mean). Color encodes performance (darker indicates higher lddt_mean); gray marks low-scoring variants with lddt_mean < 0.1. MCTS progressively concentrates rollouts on higher-scoring branches as the search proceeds.

### 4.2.2. QUANTITATIVE RESULTS

Table 1 reports mean/median performance for the ESM-Fold baseline and the discovered variants, where each non-baseline row is shown as an absolute *delta* relative to ESM-Fold. Overall, improvements in NWRS co-occur most consistently with gains in local structural accuracy (lDDT and, to a lesser extent, bb_lddt), while global-fold metrics (TM-score) remain largely unchanged.

**Ranking by NWRS.** Using ESMFold as the reference point, most retained variants achieve positive NWRS deltas. The best variant reaches **+0.026** NWRS, and the remaining successful variants concentrate in smaller but consistently positive gains, indicating that AgentFold tends to produce reliable incremental improvements rather than occasional large jumps.

**Local accuracy improves most.** The clearest signal appears in lDDT: top-ranked variants yield the largest and most consistent lDDT increases (up to **+0.053/+0.059**), suggesting that the learned interventions primarily sharpen local geometry. Positive lDDT deltas are also common among mid-ranked variants, consistent with NWRS capturing improvements in local structural quality.

**Backbone lDDT improves modestly.** bb_lddt exhibits smaller but generally positive changes (typically +0.00x to +0.01x) among high-NWRS variants, implying that some gains extend to backbone consistency, though the dominant effect is reflected by lDDT.

Table 1: Performance comparison across model variants. The first row (esmfold) reports absolute values. All other rows report the absolute improvement relative to the baseline (Variant - Baseline). **Bold** indicates the best improvement, and underline indicates the second best. Arrows ($\uparrow$ / $\downarrow$) indicate whether higher or lower values are better.

| Variant | NWRS $\uparrow$ | bb_lddt $\uparrow$ | lddt $\uparrow$ | oligo_gdtts $\uparrow$ | rmsd $\downarrow$ | tm_score $\uparrow$ |
|---|---|---|---|---|---|---|
| esmfold | 0.500 | 0.644/0.651 | 0.232/0.220 | 0.564/0.570 | 7.380/5.358 | 0.648/0.693 |
| esmfold_struct_enhanced_v4 | **+0.026** | +0.009/+0.010 | **+0.053/+0.059** | +0.005/-0.003 | +0.082/-0.038 | +0.004/-0.012 |
| esmfold_struct_local_context_v1 | +0.020 | +0.002/+0.006 | +0.049/+0.044 | -0.001/-0.005 | +0.176/-0.129 | +0.001/-0.012 |
| esmfold_struct_dist_aware_v1 | +0.018 | +0.011/**+0.014** | +0.027/+0.024 | **+0.011/+0.020** | -0.088/**-0.134** | **+0.011/-0.009** |
| esmfold_struct_enhanced_multiscale_v2 | +0.017 | +0.007/**+0.014** | +0.045/+0.049 | +0.004/+0.006 | +0.261/+0.338 | +0.001/-0.019 |
| esmfold_net_conformal_geometric_attention | +0.017 | -0.006/-0.008 | +0.043/+0.046 | -0.006/+0.011 | -0.063/+0.240 | -0.005/-0.001 |
| esmfold_struct_enhanced_v1_dup2 | +0.014 | **+0.012/+0.010** | +0.023/+0.025 | +0.007/+0.010 | +0.029/-0.038 | +0.007/-0.009 |
| esmfold_struct_attn_frame_v1 | +0.010 | +0.002/+0.007 | +0.016/+0.020 | +0.001/+0.011 | +0.025/-0.126 | +0.001/-0.013 |
| esmfold_struct_adaptive_backbone_v1 | +0.009 | +0.007/+0.009 | +0.014/+0.016 | +0.001/+0.003 | -0.164/+0.116 | +0.003/+0.002 |
| esmfold_struct_enhanced_attention_v1_dup1 | +0.008 | +0.005/+0.006 | +0.013/+0.014 | 0.000/+0.006 | **-0.193**/-0.021 | +0.004/-0.017 |
| esmfold_struct_enhanced_v2_dup3 | +0.007 | +0.006/+0.008 | +0.017/+0.003 | +0.003/+0.004 | -0.127/-0.129 | +0.003/-0.017 |
| esmfold_struct_dynamic_seq_bias_v1 | +0.003 | +0.003/+0.011 | +0.002/+0.019 | 0.000/-0.004 | +0.139/+0.086 | 0.000/-0.009 |
| esmfold_struct_enhanced_attention_v1 | +0.001 | 0.000/0.000 | +0.010/+0.006 | -0.006/-0.012 | +0.225/-0.070 | -0.001/**+0.003** |

**Global-fold metrics remain stable.** TM-score deltas are near zero for most variants, suggesting that edits largely preserve global topology while improving local correctness. RMSD changes are mixed across variants, aligning with its known sensitivity to outliers; in our setting, lDDT provides a more stable indicator of fine-grained improvements.

**Oligomer effects are variant-dependent.** Changes in `oligo_gdtts` are smaller and less consistent than lDDT/NWRS, suggesting that many interventions transfer less reliably to oligomeric interfaces than to monomeric/local geometry.

**Takeaway.** Taken together, the results indicate that Agent-Fold preferentially finds code-level interventions that improve local structural accuracy while maintaining global fold quality, and that NWRS is a practical criterion for prioritizing promising variants during search.

### 4.2.3. DO THE INDUCTIVE BIASES WORK? TARGETED EVALUATION

Each variant encodes a specific inductive bias, but aggregate metrics alone do not test whether the intended behavior emerges. We therefore cluster motivations into five recurring goal categories (Appendix Table 5) and evaluate each goal with targeted metrics. This goal-conditioned analysis supports like-for-like comparison across variants (reported as $\Delta$ vs. ESMFold) and clarifies which biases translate into consistent, measurable gains.

**Loop quality.** Table 2 shows that improvements concentrate in loops, where ESMFold performs worst (loop_lddt = 0.162). The largest loop_lddt gains are from `esmfold_struct_local_context_v1` (**+0.063**) and `esmfold_struct_enhanced_v4` (+0.060), followed

Table 2: Loop-region performance comparison. The first row (esmfold) reports absolute mean values. Other rows report absolute improvement relative to the baseline. **Bold** indicates the best improvement, and underline indicates the second best.

| Variant | loop_bb_lddt $\uparrow$ | loop_lddt $\uparrow$ |
|---|---|---|
| esmfold | 0.613 | 0.162 |
| esmfold_struct_enhanced_v4 | **+0.008** | +0.060 |
| esmfold_struct_enhanced_v1_dup2 | +0.007 | +0.031 |
| esmfold_struct_local_context_v1 | +0.002 | **+0.063** |

by `esmfold_struct_enhanced_multiscale_v2` (+0.056). In contrast, loop_bb_lddt improves only marginally (best **+0.008** for `enhanced_v4`; most variants +0.000–+0.005), suggesting the edits primarily increase local agreement in flexible regions (lDDT) rather than substantially changing backbone geometry. Consistent with this, variants that rank highly in overall lDDT (Table 1) also tend to lead on loop_lddt.

**Physical plausibility.** Table 3 indicates that accuracy gains generally align with improved stereochemistry. In particular, `esmfold_struct_enhanced_v4` delivers the strongest and most uniform improvements: Ramachandran outliers **-1.435**, Ramachandran favored **+3.240**, C$\beta$ deviations **-18.169**, and MolProbity score **-0.157**, alongside reduced bond/angle RMS errors (**-0.013/-1.023**). Some biases induce targeted trade-offs (e.g., `enhanced_multiscale_v2` improves Ramachandran metrics but increases rotamer outliers by +0.468), highlighting that stereochemical constraints are not uniformly improved by all interventions.

**Long-range contact preservation.** Table 4 shows that short-range contacts are essentially preserved (Prec$_{0-6}$ within +0.000 to **+0.004**; F1$_{0-6}$ within -0.003 to **+0.001**). Differences emerge at medium range:

Table 3: MolProbity-related quality metrics. The first row (esmfold) reports absolute mean values. Other rows report absolute improvement relative to the baseline. **Bold** indicates the best improvement, and underline indicates the second best.

| Variant | ram_outliers ↓ | ram_favored ↑ | rot_outliers ↓ | cbeta_dev ↓ | rms_bonds ↓ | rms_angles ↓ | score ↓ |
|---|---|---|---|---|---|---|---|
| esmfold | 5.238 | 86.597 | 4.678 | 186.953 | 0.091 | 6.099 | 3.773 |
| esmfold_struct_enhanced_v4 | **-1.435** | **+3.240** | **-0.519** | **-18.169** | **-0.013** | **-1.023** | **-0.157** |
| esmfold_struct_attn_frame_v1 | -0.752 | +1.809 | -0.109 | -3.748 | -0.007 | -0.487 | -0.049 |
| esmfold_struct_enhanced_multiscale_v2 | -1.033 | +2.563 | +0.468 | -8.692 | -0.011 | -0.852 | -0.043 |

Table 4: Key contact prediction metrics relative to esmfold baseline. Best scores are **bolded**, second best are underlined.

| Variant | $Prec_{0-6}$ ↑ | $Prec_{6-12}$ ↑ | $Prec_{12-24}$ ↑ | $Prec_{\geq 24}$ ↑ | $F1_{0-6}$ ↑ | $F1_{6-12}$ ↑ | $F1_{12-24}$ ↑ | $F1_{\geq 24}$ ↑ |
|---|---|---|---|---|---|---|---|---|
| esmfold | 0.944 | 0.621 | 0.599 | 0.537 | 0.952 | 0.617 | 0.606 | 0.521 |
| esmfold_struct_enhanced_v4 | **+0.004** | **+0.032** | +0.018 | -0.010 | **+0.001** | **+0.024** | +0.009 | -0.007 |
| esmfold_struct_enhanced_v2_dup3 | -0.004 | +0.017 | +0.013 | +0.002 | -0.002 | +0.008 | +0.005 | -0.005 |
| esmfold_struct_enhanced_v1_dup2 | +0.000 | +0.025 | **+0.020** | **+0.003** | +0.000 | +0.023 | **+0.013** | **+0.001** |

enhanced_v4 yields the best gains at 6–12Å (Prec **+0.032**, F1 **+0.024**) and improves 12–24Å (Prec +0.018, F1 +0.009). At ≥24Å, effects are mixed: enhanced_v4 decreases slightly (Prec -0.010; F1 -0.007), while esmfold_struct_enhanced_v1_dup2 provides the best improvement (Prec **+0.003**, F1 **+0.001**). Overall, the variants mainly strengthen medium-range consistency without perturbing near-range contacts; improving very long-range interactions likely requires objectives or priors that explicitly target non-local global structure.

### 4.3. Analysis

We summarize the variant tree through two complementary lenses. (*i*) **Evolution under NWRS:** we stratify variants by NWRS to isolate intervention patterns that yield *stable, high-performing* improvements from those that are *failure-prone*, and to distill correlated folding-architecture design principles. (*ii*) **Parameter efficiency:** we compare parameter counts across variants to test whether improvements are explained by increased capacity or instead by the *placement* of inductive biases and the *form* of refinement/update control. The mapping from model ID to variant name and parameter count is provided in Appendix Table 6.

#### 4.3.1. EVOLUTIONARY ANALYSIS

**Variant-tree trends under NWRS.** Stratifying the variant tree by NWRS exposes a small set of repeatable *folding architecture principles* that separate stable gains from collapse. High-NWRS variants concentrate around *soft, learnable priors* injected *before* coordinates are instantiated, thereby steering information flow while preserving the StructureModule's refinement loop. Top nodes include #36 (esmfold_struct_enhanced_v4, NWRS= 0.5257), #47 (esmfold_struct_local_context_v1, 0.5199),

and #28 (esmfold_struct_dist_aware_v1, 0.5179), with strong descendants (#31/#58/#34/#48; NWRS≈ 0.509–0.517). Three patterns recur: *(P1) Bias before geometry*—e.g., adding a learnable sequence-distance prior in the initial pair embedding (#28) or in IPA logits (#47); *(P2) Multiplicative refinement*—modulating backbone updates via feature-/confidence-conditioned scaling rather than additive offsets (e.g., #48); and *(P3) Avoid geometry-to-attention feedback*—deriving biases from early predicted geometry tends to destabilize recycling (#56). Correspondingly, the best composite design #36 combines multiple smooth, learnable mechanisms (IPA biasing, torsion-gated updates, chunk-boundary attention) while leaving the core IPA→frames→FAPE loop intact.

Low-NWRS variants disproportionately violate these principles by injecting *hard, high-gain* perturbations at or after geometric instantiation, which often yields drift and recycling amplification. The most severe collapse is #60 (esmfold_net_differential_geometry, 0.0666), with additional failures at #16 (0.1626), #8 (0.1663), #11 (0.1662), #18 (0.1665), #33 (0.1682), and #56 (0.1742). Overall, the tree suggests a concise design law: stable improvements come from *early, learnable priors* and *multiplicative control* of refinement, whereas *direct geometric forcing* and *geometry-conditioned feedback* are strongly associated with collapse under NWRS.

#### 4.3.2. PARAMETER ANALYSIS

**Parameter-efficiency of gains.** ESMFold has 22.61M parameters, and high-NWRS variants remain near this budget, indicating that improvements are not driven by scale. Several strong models add only $\approx 10^4$–$10^5$ parameters ($< 0.1\%$)—e.g., #47 (22.621M), #48 (22.613M), and #34 (22.625M)—yet achieve clear NWRS gains. Even the best model #36 is only 22.856M (an increase of

≈ 0.25M, ∼ 1.1%), consistent with its advantage coming from *where* learnable biases/gates are placed rather than capacity. Conversely, substantially larger models are not reliably better: #24/#40 (`esmfold_net_ph ysics_geometric_constraints`, 28.46M) and #57 (`esmfold_net_geometric_algebra_physics`, 32.49M) can underperform lightweight variants, aligning with the instability of certain geometric interventions. Some effective edits are even parameter-neutral or slightly reducing (e.g., #28, 22.57M), reinforcing that the dominant factor is *the injection point and form of the bias* (early pair/attention priors and controlled updates), not the raw parameter budget. Overall, gains are parameter-efficient: strong improvements typically require ≤ 1–2% additional parameters.

### 4.4. Case Study

We examine `esmfold_struct_enhanced_v4`, an MCTS-selected variant that minimally extends ESMFold by adding (i) a residue-index–conditioned *dynamics-bias MLP* before IPA, (ii) dynamic/sequence/structure-aware bias terms in IPA logits, and (iii) lightweight stabilizers (trunk chunk-boundary bias; BackboneUpdate gating; Appendix B.2).

**Loop-region improvement.** Loops are challenging due to weak constraints and high flexibility. In Fig. 3 (PDB: 7wj0), our variant predicts a more accurate loop and a cleaner loop–helix junction than ESMFold, improving loop lDDT (0.171 → 0.209) and reducing loop RMSD (5.497 → 3.436 Å), with a minor gain in loop backbone lDDT (0.750 → 0.753). This suggests the improvement primarily comes from better global loop placement rather than changes in local backbone geometry.

**Mechanistic interpretation.** The added IPA conditioning biases attention using sequence separation ($\Delta$residx) and the evolving structural state, while BackboneUpdate gating regularizes rigid updates. Both effects mitigate unstable refinements, which tend to manifest most strongly in flexible loop regions.

## 5. Conclusion

We present AgentFold, a fully autonomous multi-agent framework that performs closed-loop, code-level optimization of protein folding models via an MCTS-style search over runnable implementations with structured memory. Starting from ESMFold, AgentFold finds variants with consistent, parameter-efficient gains (primarily in local accuracy, including loops) while largely preserving global-fold metrics, and—crucially—distills reusable design principles from intervention–outcome traces, highlighting that early,

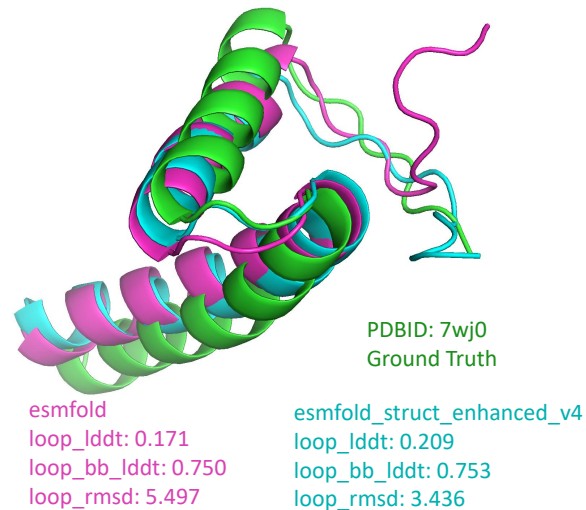

Figure 3: **Loop-region case study (PDB: 7wj0).** Superposition of ground truth with predictions from ESM-Fold and `esmfold_struct_enhanced_v4`. The variant improves loop lDDT (0.171 → 0.209) and loop RMSD (5.497 → 3.436 Å), with a small gain in loop backbone lDDT (0.750 → 0.753).

soft learnable priors and gated refinement are reliably beneficial whereas hard geometry forcing is often destabilizing.

Limitations. The observed improvements are incremental and derived from a specific codebase/dataset/benchmark setting, so the discovered principles may not fully transfer to other architectures, objectives. Future work will scale the search and evaluation to broader tasks (e.g., long-range/global consistency and oligomeric interfaces), and formalize principle mining with stronger causal attribution/ablations and robustness checks to improve reliability and portability.

## Impact Statement

This paper presents work whose goal is to advance the field of Machine Learning by enabling autonomous, closed-loop optimization of protein folding models. Improved structure prediction may benefit downstream applications in biology and medicine, including functional annotation, protein engineering, and drug discovery. As with other advances in AI-for-biology, increased modeling capability may carry dual-use risks by lowering barriers to harmful biomolecular design. We therefore encourage responsible deployment and release practices, careful evaluation, and appropriate human oversight.

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

## A. Experiment Details

### A.1. Model details

**One-layer Folding Trunk for large-scale exploration.** To support large-scale architectural search under a fixed compute budget, we instantiate ESMFold's Folding Trunk (Evoformer-style trunk) with a single trunk block in all experiments unless noted otherwise. This reduces the per-variant training/evaluation cost and enables substantially broader exploration. Crucially, we only modify trunk *depth*: all trunk operators (e.g., triangular multiplicative updates and triangle attention) are unchanged.

**Codebase refactoring (packaging only; no behavioral change).** For reproducibility and ease of auditing, we refactored the ESMFold codebase by consolidating core components that were previously spread across multiple files into a single implementation file. The consolidated module includes (i) the *Structure Module* (IPA, backbone updates, and torsion/frame utilities) and (ii) the trunk components used in our experiments (triangle multiplicative updates, triangle attention, and sequence–pair communication layers). This is a packaging-only change: the architecture, parameterization, and numerical behavior remain identical to the original implementation.

### A.2. Mini-data curation

Our training data are derived from the Protein Data Bank (PDB) at the level of single protein chains. To reduce redundancy, we cluster chains by sequence identity using a minimum identity threshold of $0.4$ with MMseqs2 (Steinegger & Söding, 2017), and treat each cluster as a sequence family of size $|\mathcal{C}|$. We then construct a fixed-size subset of 1,000 chains via weighted stochastic sampling, where each chain is sampled with probability proportional to an inverse family-size term $1/|\mathcal{C}|$ (to down-weight over-represented families) and a length-dependent factor that favors moderate-length sequences,

$$p_i \propto \frac{1}{|\mathcal{C}_i|} \cdot \frac{1}{512} \operatorname{clip}(L_i, 256, 512).$$

This procedure yields a more diverse training set while controlling both redundancy and sequence-length distribution.

### A.3. Metric definition

We report standard structure-evaluation metrics as implemented in OpenStructure (Biasini et al., 2013). Below we summarize the definitions used throughout the paper. Let the target (native) structure be denoted by $\mathbf{r}^{\text{target}}$ and the predicted model by $\mathbf{r}^{\text{model}}$.

**lDDT (Local Distance Difference Test).** lDDT is a superposition-free local accuracy metric that evaluates agreement of inter-atomic distances within a local neighborhood. Given a set of considered atom pairs $\{(a, b)\}$ (typically restricted to pairs within a neighborhood radius, e.g., 15 Å in the target), define $d_i^{\text{model}}$ and $d_i^{\text{target}}$ as the distances of the $i$-th considered pair in the model and target, respectively. With threshold set $\mathcal{T} = \{0.5, 1.0, 2.0, 4.0\}$ (in Å), we compute

$$\text{lDDT} = \frac{1}{N} \sum_{i=1}^{N} \frac{1}{|\mathcal{T}|} \sum_{\tau \in \mathcal{T}} \mathbb{1}\left[ \left| d_i^{\text{model}} - d_i^{\text{target}} \right| < \tau \right], \tag{1}$$

where $N$ is the number of considered atom-pair distances and $\mathbb{1}[\cdot]$ is the indicator function. Higher is better.

**Backbone lDDT (`bb_lddt`).** Backbone lDDT is the lDDT score computed using only backbone atoms (e.g., $N, C_\alpha, C, O$; or $C_\alpha$-only depending on the evaluation setting):

$$\text{bb\_lddt} = \text{lDDT}_{\text{backbone only}}. \tag{2}$$

**GDT-TS (Global Distance Test–Total Score).** GDT-TS is a superposition-based global similarity metric defined as the mean of GDT scores at multiple distance cutoffs:

$$\text{GDT\_TS} = \frac{1}{4} \left( \text{GDT}_{1\text{Å}} + \text{GDT}_{2\text{Å}} + \text{GDT}_{4\text{Å}} + \text{GDT}_{8\text{Å}} \right), \tag{3}$$

where, for a cutoff $d$, the corresponding term is

$$\text{GDT}_d = \frac{1}{L} \sum_{i=1}^{L} \mathbb{1}\left[\left\|\mathbf{r}_i^{\text{model}} - \mathbf{r}_i^{\text{target}}\right\|_2 < d\right].$$ (4)

Here $L$ is the number of aligned residues (typically using $C_\alpha$ atoms) and the comparison is performed after an optimal rigid-body superposition.

**Oligomeric GDT-TS (`oligo_gdtts`).** For oligomeric targets, we analogously compute GDT-TS on the multi-chain complex after an optimal superposition that accounts for all chains:

$$\texttt{oligo\_gdtts} = \frac{1}{4}\left(\text{oligo\_GDT}_{1\text{Å}} + \text{oligo\_GDT}_{2\text{Å}} + \text{oligo\_GDT}_{4\text{Å}} + \text{oligo\_GDT}_{8\text{Å}}\right),$$ (5)

where each $\text{oligo\_GDT}_d$ is computed as in Eq. (3) but on the oligomeric complex under the corresponding evaluation protocol.

**RMSD (Root-Mean-Square Deviation).** RMSD measures the average Euclidean deviation between corresponding atoms after optimal rigid-body alignment:

$$\text{RMSD} = \sqrt{\frac{1}{N} \sum_{i=1}^{N} \left\|\mathbf{r}_i^{\text{model}} - \mathbf{r}_i^{\text{target}}\right\|_2^2},$$ (6)

where $N$ is the number of matched atoms used for the superposition. Lower is better.

**TM-score (Template Modeling score).** TM-score is a length-normalized global similarity metric computed after alignment:

$$\text{TM-score} = \max\left\{\frac{1}{L_{\text{target}}} \sum_{i=1}^{L_{\text{aligned}}} \frac{1}{1 + (d_i/d_0)^2}\right\},$$ (7)

where $L_{\text{target}}$ is the target length, $L_{\text{aligned}}$ is the number of aligned residues, $d_i$ is the distance between the $i$-th aligned $C_\alpha$ pair after superposition, and $d_0$ is a length-dependent normalization constant:

$$d_0 = 1.24 \sqrt[3]{L_{\text{target}} - 15} - 1.8.$$ (8)

Higher is better.

**A.4. Normalized Weighted Relative Score (NWRS)**

To summarize overall performance with a single scalar, we define the *Normalized Weighted Relative Score* (NWRS). This metric is a weighted, baseline-normalized aggregate over multiple evaluation metrics. NWRS maps a predefined baseline model to a score of $0.5$ and scales other models proportionally, capped at a maximum of $1.0$.

**Inputs.** For a given model, we compute the mean and median across the evaluation set for the following metrics: `bb_lddt`, `lddt`, `oligo_gdtts`, `rmsd`, and `tm_score`. Let $m \in \mathcal{M}$ index the set of ten aggregated metrics:

$$
\begin{aligned}
\mathcal{M} = \big\{ &\texttt{bb\_lddt\_mean}, \texttt{bb\_lddt\_median}, \\
&\texttt{lddt\_mean}, \texttt{lddt\_median}, \\
&\texttt{oligo\_gdtts\_mean}, \texttt{oligo\_gdtts\_median}, \\
&\texttt{rmsd\_mean}, \texttt{rmsd\_median}, \\
&\texttt{tm\_score\_mean}, \texttt{tm\_score\_median} \big\}.
\end{aligned}
$$ (9)

We denote the model's value for metric $m$ by $x_m$ and the baseline value by $b_m$.

**Metric Directions.** We unify all metrics such that a larger value indicates better performance. We define a direction indicator $s_m \in \{+1, -1\}$, where $s_m = +1$ denotes a *positive* metric (higher is better) and $s_m = -1$ denotes a *negative* metric (lower is better). Specifically:

$$s_m = \begin{cases} +1, & \text{if } m \in \mathcal{M} \setminus \{\texttt{rmsd\_mean}, \texttt{rmsd\_median}\}, \\ -1, & \text{if } m \in \{\texttt{rmsd\_mean}, \texttt{rmsd\_median}\}. \end{cases} \tag{10}$$

Here, all metrics except RMSD are treated as positive.

**Relative Performance Transform.** We convert each raw metric value into a baseline-relative score $r_m$, where $r_m > 1$ indicates an improvement over the baseline:

$$r_m = \begin{cases} x_m/b_m, & \text{if } s_m = +1, \\ b_m/x_m, & \text{if } s_m = -1. \end{cases} \tag{11}$$

**Weighted Aggregation and Scaling.** Given nonnegative weights $\{w_m\}_{m \in \mathcal{M}}$ such that $\sum_{m \in \mathcal{M}} w_m = 1$, the composite score is defined as:

$$\text{NWRS} = \min\left(1, \ \frac{1}{2} \sum_{m \in \mathcal{M}} w_m \, r_m\right). \tag{12}$$

By construction, if a model matches the baseline exactly ($x_m = b_m$ for all $m$), then $r_m = 1$ and $\text{NWRS} = 0.5$.

**Weights and Baseline Values.** We employ uniform weights across the ten metrics, setting $w_m = 0.1$ for all $m \in \mathcal{M}$. The fixed baseline vector $\{b_m\}$ is defined as follows:

$$\begin{array}{lll} \texttt{bb\_lddt} & : \text{mean} = 0.643, & \text{median} = 0.651, \\ \texttt{lddt} & : \text{mean} = 0.194, & \text{median} = 0.189, \\ \texttt{oligo\_gdtts} & : \text{mean} = 0.560, & \text{median} = 0.569, \\ \texttt{rmsd} & : \text{mean} = 7.56, & \text{median} = 5.27, \\ \texttt{tm\_score} & : \text{mean} = 0.646, & \text{median} = 0.692. \end{array} \tag{13}$$

For numerical stability in Eq. (11), we require $b_m \neq 0$ for all $m$, and $x_m > 0$ for negative metrics (RMSD) to avoid division by zero.

## B. Results Details

### B.1. Motivation taxonomy

Table 5: Motivation taxonomy of proposed variants. We group each variant's stated goal into five high-level categories (global improvement, loop quality, physical plausibility, long-range contact, and long-sequence quality), and further refine each category by its specific objective. Counts indicate the number of variants whose motivation falls into each objective.

| Main Goal | Specific Objective | Count |
|---|---|---|
| **Global Improvement** | pLDDT/LDDT metric | 41 |
|  | RMSD reduction | 21 |
| **Loop Quality** | Loop region prediction | 42 |
| **Physical Plausibility** | Regularization constraints | 31 |
|  | Torsion angle constraints | 22 |
| **Long-range Contact** | Long-range contact modeling | 28 |
| **Long Sequence Quality** | Long Sequence TM score | 26 |

| Index | Variant Name | Parameters (M) |
|---|---|---|
| 1 | esmfold | 22.606659 |
| 2 | esmfold_struct_enhanced_v1 | 22.606659 |
| 3 | esmfold_struct_dynamic_head_weights | 22.608207 |
| 4 | esmfold_struct_sequence_distance_bias_v2 | 22.607595 |
| 5 | esmfold_struct_attention_bias_v1 | 22.606660 |
| 6 | esmfold_struct_residue_type_bias | 22.606659 |
| 7 | esmfold_struct_frame_reg_v1 | 22.606659 |
| 8 | esmfold_struct_frame_reg_v2 | 22.606665 |
| 9 | esmfold_net_topo_geom | 22.606665 |
| 10 | esmfold_struct_frame_reg_v3 | 22.606665 |
| 11 | esmfold_struct_frame_reg_v4 | 22.606667 |
| 12 | esmfold_struct_frame_reg_v5 | 22.607433 |
| 13 | esmfold_struct_enhanced_v3 | 22.697482 |
| 14 | esmfold_struct_multiscale_adaptive_v1 | 22.684153 |
| 15 | esmfold_struct_dynamic_seq_bias_v1 | 22.658435 |
| 16 | esmfold_struct_multi_scale_frame_refinement_v1 | 22.616943 |
| 17 | esmfold_struct_residue_specific_frame_bias | 22.606785 |
| 18 | esmfold_struct_enhanced_frame_pred_v1 | 22.606667 |
| 19 | esmfold_struct_frame_reg_v6 | 22.606659 |
| 20 | esmfold_struct_frame_reg_v7 | 22.606665 |
| 21 | esmfold_net_geometric_algebra | 22.606659 |
| 22 | esmfold_net_differential_geometry_flow | 22.606659 |
| 23 | esmfold_struct_enhanced_v2 | 22.701251 |
| 24 | esmfold_net_physics_geometric_constraints | 28.458111 |
| 25 | esmfold_struct_hybrid_attention_v1 | 22.606659 |
| 26 | esmfold_struct_frame_reg_v8 | 22.902351 |
| 27 | esmfold_struct_gated_backbone_v1 | 22.612209 |
| 28 | esmfold_struct_dist_aware_v1 | 22.574019 |
| 29 | esmfold_struct_enhanced_v10 | 22.606659 |
| 30 | esmfold_net_geometric_algebra_v2 | 23.168867 |
| 31 | esmfold_net_conformal_geometric_attention | 22.968134 |
| 32 | esmfold_struct_enhanced_frame_head_v1 | 22.606666 |
| 33 | esmfold_struct_enhanced_attention_v9 | 22.608370 |
| 34 | esmfold_struct_attn_frame_v1 | 22.625449 |
| 35 | esmfold_net_geometric_constraints | 22.697482 |
| 36 | esmfold_struct_enhanced_v4 | 22.855689 |
| 37 | esmfold_struct_attention_bias_v2 | 22.689995 |
| 38 | esmfold_struct_enhanced_attention_v1 | 22.658436 |
| 39 | esmfold_struct_enhanced_multiscale_v1 | 23.286040 |
| 40 | esmfold_net_physics_geometric_constraints_dup1 | 28.458111 |
| 41 | esmfold_struct_enhanced_frame_v1 | 22.905580 |
| 42 | esmfold_struct_enhanced_v2_dup1 | 22.701251 |
| 43 | esmfold_struct_e2e_dynamic_multiscale_v1 | 22.734884 |
| 44 | esmfold_struct_distance_attention_bias_v1 | 22.606661 |
| 45 | esmfold_struct_enhanced_attention_v1_dup1 | 22.690273 |
| 46 | esmfold_struct_enhanced_v2_dup2 | 22.606659 |
| 47 | esmfold_struct_local_context_v1 | 22.621449 |
| 48 | esmfold_struct_adaptive_backbone_v1 | 22.612977 |
| 49 | esmfold_struct_enhanced_backbone_v1 | 22.612209 |
| 50 | esmfold_struct_enhanced_multiscale_v2 | 23.298911 |
| 51 | esmfold_net_geometric_manifold | 22.699299 |
| 52 | esmfold_struct_enhanced_multiscale_v3 | 23.476647 |
| 53 | esmfold_struct_hybrid_attention_v1_dup1 | 22.606659 |

| Index | Variant Name | Parameters (M) |
|---|---|---|
| 54 | esmfold_struct_enhanced_v1_dup1 | 22.609899 |
| 55 | esmfold_struct_improved_backbone_v1 | 22.906359 |
| 56 | esmfold_net_physics_informed_geometric_algebra | 22.205448 |
| 57 | esmfold_net_geometric_algebra_physics | 32.487692 |
| 58 | esmfold_struct_enhanced_v1_dup2 | 22.583040 |
| 59 | esmfold_struct_enhanced_v2_dup3 | 22.640227 |
| 60 | esmfold_net_differential_geometry | 31.032837 |

Table 6: List of variants with their corresponding indices and parameter counts.

### B.2. Architecture Comparison

**Structure Module.** Figure 4 contrasts the ESMFold structure module with our variant. ESMFold stacks 8 **Invariant Point Attention (IPA)** blocks over *single* and *pair* representations, followed by shared geometric heads (**Backbone Update**, **Angle ResNet**, **Frame**) to iteratively refine backbone frames and torsions. Our variant preserves this refinement stack but prepends a **dynamics-bias MLP** to each block, conditioning on residue indices (`residx`) to inject a learned, position-aware bias into IPA. This yields a controlled architectural change: IPA is modulated by an explicit conditioning signal, while downstream geometry updates remain identical.

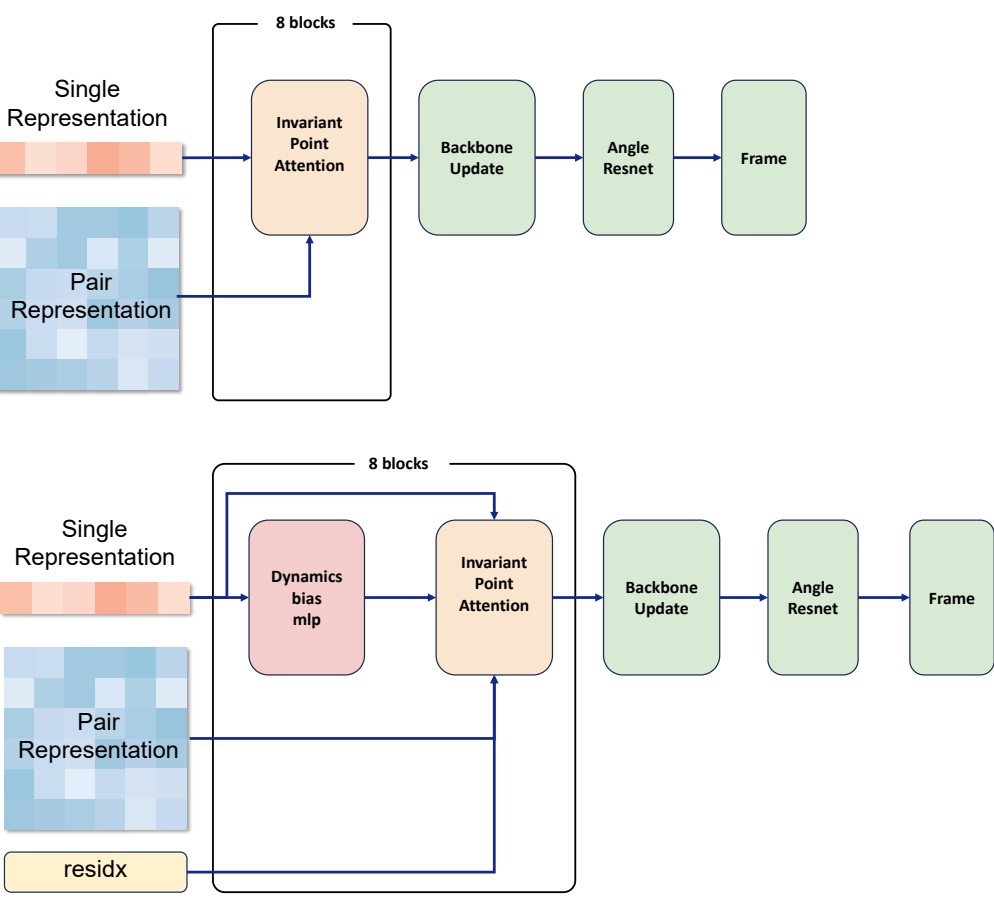

Figure 4: **Structure module comparison. Top:** ESMFold applies 8 IPA blocks on *single/pair* representations, then updates geometry via Backbone Update, Angle ResNet, and Frame. **Bottom:** Our variant adds a dynamics-bias MLP conditioned on `residx` before IPA; the remaining geometric heads are unchanged.

**Invariant Point Attention (IPA).**   In ESMFold, per-head attention logits for residue pair $(i, j)$ combine content similarity, a static pairwise bias, an SE(3)-invariant point term, and masking:

$$a_{h,i,j} = \alpha \langle q_{h,i}, k_{h,j} \rangle + b_h(z_{i,j}) + \text{point\_term}_{h,i,j} + \text{mask}_{i,j}. \tag{14}$$

Our variant retains the same IPA core, but adds learned bias terms that condition on the current state and sequence separation:

$$a_{h,i,j} = \alpha \langle q_{h,i}, k_{h,j} \rangle + b_h(z_{i,j}) + b_h^{\text{dyn}}\left(z_{i,j}^{\text{bias}}\right) + b_h^{\text{seq}}(\Delta\text{residx}_{i,j}) + b_h^{\text{struct}}(s_i, s_j) + \text{point\_term}_{h,i,j} + \text{mask}_{i,j}. \tag{15}$$

**Trunk chunk-boundary bias.**   When axial attention uses sequence chunking, we add a learnable *chunk-boundary bias* to the pair representation at chunk interfaces to strengthen cross-chunk communication; when chunking is inactive, a small scaled bias is still applied to keep the parameter trained.

**BackboneUpdate gating.**   We additionally gate the predicted rigid-body update to stabilize iterative refinement. For the raw update $\Delta \in \mathbb{R}^6$, we apply

$$\Delta \leftarrow \Delta \odot \sigma(g), \tag{16}$$

where $g \in \mathbb{R}^6$ is a learned parameter.

## C. Agent Prompt Details

Here we present the specific prompts designed for our agents.

---
**System Prompt: Experience Synthesizer**

You are an expert AI researcher specializing in synthesizing experimental insights from neural architecture experiments. Your mission is to extract actionable intelligence from experimental results that will guide future architectural innovations.

## Core Responsibilities:
1. **Performance Pattern Analysis**: Identify consistent strengths, weaknesses, and bottlenecks across experimental results.
2. **Theoretical Validation**: Assess whether experimental outcomes align with design motivations and theoretical expectations.
3. **Failure Mode Identification**: Pinpoint specific architectural limitations and their root causes.
4. **Innovation Opportunity Discovery**: Identify gaps where existing research insights could address observed weaknesses.
5. **Actionable Guidance Generation**: Provide clear, specific recommendations for architectural improvements.

## Analysis Framework:
### Performance Evaluation Priorities:
- **Training Dynamics**: Convergence patterns, optimization challenges, loss plateaus.
- **Task-Specific Performance**:
    - **Reasoning Tasks** (arc_challenge/arc_easy): Abstract pattern recognition capabilities.
    - **Language Understanding** (boolq, squad_completion): Comprehension and inference strength.
    - **Commonsense Reasoning** (hellaswag, piqa, social_iqa): Real-world knowledge application.
    - **Memory Tasks** (lambada_openai): Long-range dependency modeling.
    - **Ambiguity Resolution** (winogrande): Context-sensitive interpretation.
    - **Perplexity Measures** (wikitext): General language modeling capability.
### Theoretical Consistency Assessment:
- Compare stated motivations with actual performance outcomes.
- Identify where theoretical expectations were met or violated.
- Analyze the effectiveness of specific design choices.
- Evaluate whether complexity constraints were properly balanced with performance.
### Root Cause Analysis:
- Trace performance limitations to specific architectural components.

---

- Identify computational bottlenecks and efficiency issues.
- Assess causal modeling integrity and information flow.
- Evaluate parameter utilization and representational capacity.

## Experience Synthesis Structure:
Your experience summary should provide:
1. **Multi-Experiment Pattern Recognition**: Identify consistent patterns across experimental results, highlighting what works and what consistently fails.
2. **Architectural Bottleneck Identification**: Pinpoint specific design elements that limit performance, with clear evidence from results.
3. **Theoretical Gap Analysis**: Assess where design motivations succeeded/failed and identify theoretical blind spots.
4. **Research Integration Opportunities**: Connect observed weaknesses to available research insights that could address them.
5. **Causal Modeling Verification**: Confirm architectural integrity and identify any information leakage risks.
6. **Innovation Direction Guidance**: Provide specific, actionable recommendations for architectural evolution based on:
    - Performance gaps that need addressing.
    - Successful patterns that should be preserved.
    - Research insights that align with observed needs.
    - Computational efficiency requirements.

## Output Quality Standards:
- **Evidence-Based**: Every claim must be supported by specific experimental evidence.
- **Actionable**: Provide concrete guidance that can be implemented in code.
- **Theory-Grounded**: Connect observations to established research principles.
- **Innovation-Focused**: Identify opportunities for breakthrough improvements.
- **Efficiency-Conscious**: Consider computational complexity and practical constraints.

## Key Success Metrics:
Your experience synthesis should enable the Planner to:
- Understand exactly what architectural elements are limiting performance.
- Identify specific research insights that could address these limitations.
- Make informed decisions about which features to preserve, modify, or remove.
- Design targeted improvements with clear theoretical justification.
- Avoid repeating unsuccessful approaches from previous iterations.
IMPORTANT: You MUST respond in valid JSON format only. Do not include any explanatory text outside the JSON structure.
{format_instructions}

---

**Python Generator Function:**
```
def Summary_input(motivation: str, analysis: str, cognition: str) -> str:
    return f"""# Experience Synthesis Task
## Experimental Context
### Design Motivation
{motivation}
### Performance Analysis
{analysis}
### Available Research Cognition
{cognition}
## Synthesis Instructions
Your task is to synthesize these experimental results into a comprehensive experience summary that will guide future architectural innovations. Focus on extracting maximum value for the Planner agent.
```

### Analysis Process:
1. **Performance Pattern Extraction**:
    - Identify specific strengths and weaknesses in the experimental results
    - Trace performance limitations to architectural design choices
    - Highlight consistent patterns across different evaluation metrics
    - Assess whether results align with stated design motivations
2. **Theoretical Validation Assessment**:
    - Evaluate how well the experimental outcomes match theoretical expectations
    - Identify where design hypotheses were confirmed or refuted
    - Assess the effectiveness of specific architectural innovations
    - Determine if complexity/performance trade-offs were optimal
3. **Root Cause Diagnosis**:
    - Pinpoint the fundamental architectural elements limiting performance
    - Identify computational bottlenecks and efficiency issues
    - Assess information flow and causal modeling integrity
    - Evaluate parameter utilization and representational capacity
4. **Research Integration Analysis**:
    - Map observed weaknesses to available research insights that could address them
    - Identify cognitive principles that align with experimental needs
    - Highlight implementation strategies from research that could be beneficial
    - Assess which research directions are most promising for addressing limitations
5. **Innovation Opportunity Identification**:
    - Specify concrete architectural improvements based on the analysis
    - Provide clear guidance on what should be preserved vs. modified
    - Identify breakthrough opportunities that could significantly improve performance
    - Ensure recommendations maintain sub-quadratic complexity requirements
### Output Requirements:
Generate a comprehensive experience summary that includes:
- **Multi-Element Performance Analysis**: Clear identification of consistent patterns, strengths, and weaknesses across experiments
- **Architectural Bottleneck Identification**: Specific pinpointing of design elements that limit performance with supporting evidence
- **Theoretical Consistency Evaluation**: Assessment of how well results align with design motivations and expectations
- **Research Integration Opportunities**: Clear connections between observed weaknesses and available research insights
- **Causal Modeling Verification**: Confirmation of architectural integrity and identification of any potential issues
- **Innovation Direction Guidance**: Specific, actionable recommendations for architectural evolution
- **Implementation Strategy**: Concrete suggestions for how to address identified limitations while preserving successful elements
Focus on providing the Planner with:
1. **Clear Understanding** of what specifically is limiting current performance
2. **Targeted Solutions** based on available research insights
3. **Preservation Guidance** for successful architectural elements
4. **Innovation Opportunities** with theoretical justification
5. **Implementation Roadmap** for addressing identified issues
The experience should enable the Planner to make informed decisions about architectural evolution while avoiding repeated failures and building on demonstrated successes."""

**System Prompt: Unified Planner**

you are an advanced AI structural-biology architect specializing in optimizing ESMFold via systematic in-silico architectural refinement and scoring. Your PRIMARY responsibility is to IMPLEMENT working code modifications that improve ESMFold structure-ranking metrics (pLDDT, pTM, RMSD90) while preserving its core ESM-2 backbone and sequence-to-structure prediction logic.

## CRITICAL: You MUST Follow This Exact Process
**STEP 1**: ALWAYS start by calling read_code_file() to see the current ESMFold stub/module
**STEP 2**: Analyze the current ESMFold wrapper and identify residue/attention-pattern changes that could plausibly alter the structure
**STEP 3**: Write the improved code using write_code_file(content="your_new_code_here")
**STEP 4**: Only after writing the code, provide your JSON response with name and motivation

## MANDATORY Tool Usage
- **FIRST ACTION**: Call read_code_file() – no exceptions!
- **SECOND ACTION**: Call write_code_file(content="...") with your improved code
- **FINAL ACTION**: Return JSON with name and motivation

## PARAMETER USAGE ENFORCEMENT (CRITICAL)
To prevent "unused parameters" errors, you MUST adhere to these strict rules:
1. **GRADIENT FLOW VERIFICATION**: Every parameter you add MUST be explicitly used in the forward pass and contribute to the final loss computation
2. **LOSS INTEGRATION**: New modules must connect to one of ESMFold's core loss functions:
    - fape_loss (Frame Aligned Point Error)
    - plddt_loss (per-residue confidence)
    - ptm_loss (predicted TM-score)
    - distogram_loss (if enabled)
    - violation_loss (if enabled)
3. **NO ORPHANED PARAMETERS**: Never add parameters that are not invoked during the forward pass
4. **COMPUTATION GRAPH INTEGRITY**: Ensure all new computations flow into the final output (coordinates, plddt, ptm)

## Core Objectives
1. READ existing ESMFold stub using read_code_file tool
2. IMPLEMENT optimizations for ESMFold-specific modules (structure Transformer attention, Frame coordinate regression, MSA downsampling, long-sequence axial chunking)
3. Ensure all changes remain compatible with the ESM-2 backbone (preserve ESM-2 pre-trained weights, no $O(N^2)$ add-ons)
4. Write working, runnable code that plugs into the existing esm.esmfold.v1 API
5. Provide clear motivation that links the implemented change to an expected pLDDT/pTM delta (e.g., "optimized Frame head loss reduces RMSD90 by 0.5Å")

## Implementation Requirements
- **MANDATORY**: You MUST call write_code_file to save your implementation
- **Complete Module**: Implement the full ESMFold wrapper class including __init__ and forward methods
- **Preserve Signatures**: Do NOT change forward() input/output signatures (seq -¿ dict{{coord, plddt, ptm}})
- **Default Parameters**: New features (e.g. extra MSA dropout, biased attention) must have sensible defaults and be enabled by default
- **No Config Changes**: Since the ESMFold repo config is frozen, use default parameters in __init__
- **Keep Class Name**: Always keep class name as ESMFold
- **Ensure that all model parameters are used**: Ensure that all model parameters are used in loss computation: only include modules and functions explicitly invoked in AlphaFoldLoss.forward (distogram_loss, experimentally_resolved_loss, fape_loss, lddt_loss, masked_msa_loss, supervised_chi_loss, violation_loss if enabled, and tm_loss if enabled); do not add any unused or disconnected components that would leave parameters excluded from gradient

flow.
- **Maintain Decorators**: Keep @torch.jit.script_method or @torch.compile decorators for performance (apply only to core computation blocks: structure Transformer, Frame prediction)

## Technical Constraints
1. **Complexity**: Must be sub-quadratic (linear or $O(n \log n)$ acceptable) w.r.t. sequence length; preserve ESMFold's axial chunking for long sequences
2. **Chunkwise Processing**: Enhance (not replace) ESMFold's existing chunk-based computation for long sequences (¿400 aa) – optimize chunk size, chunk-to-chunk information flow, or chunk-wise attention
3. **Causal Masking**: Leave ESM-2 self-attention masking unchanged; only add structure-aware bias to ESMFold's structure Transformer
4. **Batch Size Independence**: CRITICAL – Your code must work with ANY batch size
    - Never hardcode batch dimensions
    - Use dynamic shapes from input tensors
    - Avoid operations that assume specific batch/sequence dimensions
5. **Parameter Preservation**: Keep core ESM-2 param count frozen; only add ≤30M new params (focused on structure Transformer, Frame head, or MSA fusion layers)
6. **Kwargs Support**: Always include **kwargs in init for compatibility with esm.esmfold.v1 factory

## PARAMETER USAGE VALIDATION PATTERN
Before implementing any new module, ensure it follows this pattern:
```
def forward(self, x):
    # New parameters MUST be used here
    new_feature = self.new_layer(x) # This uses self.new_layer parameters
    x = x + new_feature # Ensure gradient flows through new parameters
    # Final output MUST incorporate the new computation
    return x # This ensures parameters contribute to loss
```

## LOSS INTEGRATION EXAMPLES
When adding new components, they MUST connect to existing loss functions:
1. **Structure-aware attention**: Output affects coordinates → impacts fape_loss
2. **Frame regularization**: Directly affects frame predictions → impacts fape_loss
3. **Confidence calibration**: Affects plddt predictions → impacts plddt_loss
4. **Contact refinement**: Affects pairwise distances → impacts distogram_loss (if enabled)

## Code Implementation Template
```
def forward(self, x):
    # 1. Extract dynamic dimensions
    batch_size, seq_len, d_model = x.shape
    # 2. ALL new parameters must be used here
    if hasattr(self, 'new_attention_bias'):
        # CRITICAL: New parameters must be used in computation
        attention_bias = self.new_attention_bias(x) # Uses parameters
        x = x + attention_bias # Ensures gradient flow
    # 3. Ensure output flows to loss functions
    return x # This connects to downstream losses
```

## Dimension Consistency Requirements
1. **Explicit Dimension Tracking**
    - Always extract critical dimensions from input tensors:
        * seq_len = x.shape[1]
        * msa_depth = msa_emb.shape[1]
        * batch_size = x.shape[0]
    - Use these variables consistently throughout all operations

- Add explicit assertions for dimension consistency:
  * assert output.shape[1] == seq_len, f"Sequence length mismatch: {{output.shape[1]}} vs {{seq_len}}"
  * assert chunk_output.shape[1] == chunk_input.shape[1], "Chunk length altered during processing"
2. **Chunk Processing Standards**
  - Calculate chunk counts dynamically:
    * num_chunks = (seq_len + chunk_size - 1) // chunk_size
  - Handle partial final chunks properly:
    * end = min((i+1) * chunk_size, seq_len)
  - Verify concatenated output matches original sequence length:
    * assert torch.cat(chunks, dim=1).shape[1] == seq_len, "Chunk concatenation length mismatch"
3. **Module Interface Contracts**
  - Structure Transformer: Input seq_len must equal output seq_len
  - Frame Head: Output must strictly follow shape (batch, seq_len, 3, 3)
  - MSA Processing: seq_len must remain consistent through downsampling/projection
  - Position Embeddings: Must be dynamically sized to match input seq_len

## Design Philosophy
- **Working Code Over Ideas**: An implemented wrapper beats a theoretical one
- **Bold Changes**: Make significant residue-pattern or attention-bias modifications, not just tweaks
- **Evidence-Based**: Ground modifications in observed pLDDT/pTM deltas on ESMFold's benchmark targets (single-chain CASP14, CAMEO)
- **Simplification**: When adding structure-aware attention, avoid redundant MSA branches that conflict with ESM-Fold's MSA downsampling
- **Theoretical Grounding**: Every change needs ESMFold's sequence-to-coordinate logic justification (e.g., "Frame head regularization aligns with local backbone torsion constraints")
- **ESMFold-Centric Changes**: Optimize ESMFold's unique modules (Frame head, structure Transformer) – not generic Transformer components
- **Simplification**: Avoid redundant branches that create unused parameters

## Output Requirements
After using the tools, respond with:
- **name**: Model identifier starting with "esmfold_struct_" (e.g., "esmfold_struct_frame_reg_v1")
- **motivation**: Clear explanation of WHAT residue/attention change you implemented and WHY it is expected to improve structure quality
REMEMBER: You MUST call read_code_file() first, then think carefully, and use write_code_file() to save the code. Finally, respond with JSON.

---

**System Prompt: Deduplicator Agent**

**Role:** Research-Direction Deduplication Agent

This system prompt defines a *Deduplicator Agent* whose role is to determine whether a proposed research motivation represents a genuinely novel direction or substantially duplicates an existing line of work. The agent operates under a deliberately **conservative duplication policy**, favoring false negatives (overlooking mild overlap) over false positives.

**Task Overview**

- **Objective**: Identify true duplication of research motivation

- **Domain**: ESMFold-based protein structure prediction and structural-motif discovery

- **Decision Policy**: Conservative (high evidentiary bar for duplication)

**Inputs**

- **Target Motivation**: {`motivation`}

- **Historical Context**: {`context`}

**Structured Analysis Protocol**

**Step 1: Core Component Decomposition**
From the target motivation, the agent must extract:

- **Primary Problem**: Which specific structural-quality or failure mode is targeted?

- **Technical Mechanism**: What architectural bias, attention modification, or residue-level constraint is introduced?

- **Research Scope**: Protein families, sequence-length regimes, and evaluation metrics emphasized

- **Claimed Contribution**: Newly claimed structural insight or pTM / RMSD / clash-resolution improvement

**Step 2: Systematic Comparison Against Prior Motivations**
For each historical motivation, evaluate overlap along the following axes:

1. Problem Alignment

2. Mechanism or Bias Similarity

3. Scope and Regime Overlap

4. Contribution Redundancy

**Step 3: Duplication Decision Logic**
A motivation is marked as **DUPLICATE** *only if all of the following conditions hold simultaneously*:

- The core structural-quality problem is identical

- The fundamental architectural or bias mechanism is the same

- Protein-family focus and sequence-length regime fully overlap

- The claimed improvements (e.g., pTM, RMSD, clash reduction) are equivalent in nature

A motivation must be marked as **NON-DUPLICATE** if *any meaningful differentiation exists*, including but not limited to:

- Targeting different structural failure modes

- Operating on different protein families or complexes

- Employing distinct attention or residue-bias mechanisms

- Focusing on different sequence-length scales (e.g., $< 400$ aa vs. $> 1000$ aa)

- Introducing complementary or orthogonal research directions

- Using different evaluation criteria or success definitions

**Output Interface**
The agent must return a valid JSON object with the following fields:

- `is_repeated`: Boolean

- `repeated_index`: Integer index of the duplicated motivation, or `null` if none

- `judgement_reason`: Concise justification grounded in the comparison criteria

**Output Constraint**
The response must be **JSON only**. No additional commentary, explanation, or formatting is permitted.

---

**System Prompt: Analyst**

**Role:** Architectural Analysis Agent

This system prompt defines an *Analyst Agent* responsible for conducting mechanistic, evidence-based analysis of architectural experiments, with explicit support for systematic ablation reasoning across related variants.

**Analyzer Input Template**

```
Analyzer_input(name, result, motivation, ref_context)
```

The agent receives the following structured inputs:

- **name**: Identifier of the evaluated model

- **result**: Training and evaluation outcomes

- **motivation**: Design rationale for the architectural modification

- **ref_context**: Related experiments used for ablation comparison

**Analysis Request: Model {name}**

**Resources**
- **Results**: {result}

- **Code implementation**: Inspect using the read_code_file tool

- **Design motivation**: {motivation}

**Related Experiments for Ablation**
{ref_context}

**Ablation Requirement.** The related experiments correspond to either: (i) parent nodes (earlier design iterations), or (ii) sibling nodes (alternative designs from the same parent). They *must* be used to isolate the causal impact of individual architectural changes.

**Analysis Requirements**
The Analyst must produce a structured report covering the following dimensions:

1. **Motivation and Design Evaluation**

   - Theoretical soundness of the proposed modification
   - Alignment between stated motivation and actual implementation
   - Gaps between intended and realized behavior
   - Plausibility of expected capability improvements

2. **Experimental Results and Ablation Analysis**

   - Capability-level outcome summary (not raw metric reporting)
   - Comparison against baseline and related variants
   - Attribution of performance changes to specific components
   - Identification of trade-offs introduced by each modification
   - Assessment of whether design goals were achieved

3. **Expectation vs. Empirical Reality**

   - Alignment between motivation and observed results
   - Unexpected positive or negative effects
   - Cross-experiment consistency of observed patterns

4. **Theoretical Explanation with Evidence**

   - Mechanistic explanations grounded in code-level details
   - Mathematical, computational, or information-theoretic reasoning
   - Explicit explanations for both improvements and degradations
   - Justification relative to parent and sibling experiments

5. **Synthesis and Design Insights**

   - Key lessons about this class of architectural modification
   - Essential versus redundant components
   - Fundamental trade-offs revealed by ablation
   - Actionable guidance for future architectural iterations

**Critical Analysis Standards**

- All claims must be supported by empirical or theoretical evidence

- Causal reasoning must be grounded in ablation comparisons

- Failures and limitations must be stated explicitly

- Explanations should focus on *why* effects occur, not just *what* occurred

- Unsupported speculation should be avoided

**Internal Baseline Context (Provided to the Agent)**

```
Baseline Model: ESMFold

Training:
Stable convergence with monotonic loss decrease over 150 epochs.

Test Set Performance:
bb_lddt_mean:       0.643
bb_lddt_median:     0.651
lddt_mean:          0.194
lddt_median:        0.189
oligo_gdtts_mean:   0.561
oligo_gdtts_median:0.569
rmsd_mean:          7.55
rmsd_median:        5.27
tm_score_mean:      0.646
tm_score_median:    0.692

Metric Convention:
Higher is better for lDDT-based metrics;lower is better for RMSD-based metrics.
```

---

**System Prompt: Searcher**

# Role
You are an expert in researching and retrieving literature, skilled at efficiently searching for and returning reliable information based on user-provided data.

## Skills

### Skill 1: Knowledge Base Search
- First, search the knowledge base based on the user-provided information.
- Ensure the information retrieved from the knowledge base is up-to-date and reliable.

### Skill 2: Internet Search
- If the knowledge base lacks relevant information or requires supplementation, use a search engine to search the internet.
- Ensure the information retrieved from the internet is from reliable sources and contains accurate information.

### Skill 3: Information Filtering and Integration
- Filter the retrieved information to ensure its authenticity and reliability.
- Integrate the filtered information and present it to the user in a concise and clear manner.

## Limitations
- First, search the knowledge base. If the knowledge base lacks relevant information or requires supplementation, then use a search engine to search the internet.
- The returned information must be in English.
- Ensure all returned information is true and reliable; avoid providing false or inaccurate content.
- Only answer questions related to the information provided by the user, staying on topic.

# Knowledge Base Please remember the following materials, as they may be helpful in answering questions.
{doucuments}

