# OpenReview forum: "AgentFold: Self-Evolving Exploration of Protein Folding Models"
_ICML.cc/2026/Conference — Submitted to ICML 2026_

### Official Review · Reviewer_FVuB · 2026-02-25

**Soundness:** 3
**Presentation:** 2
**Significance:** 2
**Originality:** 2
**Overall Recommendation:** 3
**Confidence:** 4

**Summary:**

The authors introduce AgentFold, an autonomous agentic system for researching and developing protein folding models. Empirically, the authors find that AgentFold achieves small yet notable improvements to ESMFold's ability to capture accurate local structural details of 3D protein structures, while global fold details remain largely unchanged by AgentFold's architectural modifications. Overall, this work points to a future where autonomous systems may begin to help researchers experiment with new architectural ideas in a rapid and principled manner. However, the current study appears relatively limited in scope.

**Compliance With Llm Reviewing Policy:**

Affirmed.

**Final Justification:**

The authors did not address my initial concerns about this manuscript. As such, I am comfortable giving this work a "weak reject".

**Key Questions For Authors:**

1. What are the main bottlenecks that methods like AgentFold currently face on the road towards improved performance? Is it simply about enhancing the base LLM (e.g., Gemini 3, GPT-5), or does the software tooling in AI4Science need to improve considerably?

**Limitations:**

As the authors have discussed, their method cannot achieve substantial improvements over the baseline ESMFold model's design.

**Strengths And Weaknesses:**

**Points of Strength:**

1. The authors propose the first example in AI4Science, of which I am aware, of improving neural network architecture designs using agentic systems.
2. The authors carefully study when, why, and how AgentFold achieves architectural improvements over the baseline ESMFold model.
3. Limitations of AgentFold, in terms of its ability to improve over ESMFold, are acknowledged.

**Points for Improvement:**
1. The improvements that AgentFold suggests are somewhat obvious to a domain expert. For example, featurizing noisy 3D (predicted) structures in an AlphaFold 2-style model, as AgentFold tried to do, will introduce considerable training instabilities, especially early on in training. One can see this with a few preliminary attempts at modifying ESMFold. Furthermore, I believe DeepMind may have experimented with this idea early on in the development of AlphaFold 2, as shown in their folding animations accompanying their manuscript published in Nature [1].
2. The paper offers relatively limited applicability beyond protein-specific structure prediction. Nowadays, biomolecular co-folding has largely supplanted protein-specific folding efforts [2], and the main methodologies beyond such co-folding approaches have changed considerably (compared to how ESMFold is designed).

**References:**

[1] Jumper, J., Evans, R., Pritzel, A., Green, T., Figurnov, M., Ronneberger, O., ... & Hassabis, D. (2021). Highly accurate protein structure prediction with AlphaFold. nature, 596(7873), 583-589.

[2] Abramson, J., Adler, J., Dunger, J., Evans, R., Green, T., Pritzel, A., ... & Jumper, J. M. (2024). Accurate structure prediction of biomolecular interactions with AlphaFold 3. Nature, 630(8016), 493-500.

---

### Official Review · Reviewer_73gU · 2026-02-27

**Soundness:** 3
**Presentation:** 3
**Significance:** 3
**Originality:** 3
**Overall Recommendation:** 3
**Confidence:** 3

**Summary:**

AgentFold is a multi-agent LLM framework that uses MCTS over code-level edits to autonomously improve ESMFold's architecture. The best variant out of ~80 candidates gains +0.026 NWRS and +0.053 mean lDDT on CAMEO2022, while TM-score stays flat. The authors also distill three design principles from the search trajectory, claiming soft learnable priors outperform hard geometric perturbations.

**Compliance With Llm Reviewing Policy:**

Affirmed.

**Final Justification:**

I would like to keep my score since no response is updated.

**Key Questions For Authors:**

- Appendix A.4's fixed baseline has lddt_mean=0.194, but Table 1's ESMFold baseline is 0.232. Could the authors clarify this discrepancy?

- The Experience Synthesizer prompt in Appendix C lists arc_challenge, hellaswag, piqa etc. as evaluation priorities. These are NLP benchmarks. Why is there such a domain mismatch?

- The vairant enhanced_v4 introduces four simultanous changes. Have the authors tried removing them one at a time? E.g., is BackboneUpdate gating alone sufficient for the loop lDDT gain, or does it need the IPA logit terms?

**Strengths And Weaknesses:**

## Strengths

- MCTS over runnable code snapshots is a distinctive framing compared to standard NAS over hyperparams or graphs. The authors get better model and distilled design principles.

- The evaluation is comprehensive. Targeted breakdowns by loop quality, MolProbity stereochemistry, and contact precision at four distance ranges let me see what each edit actually changes structurally, e.g. some variants improve Ramachandran metrics while increasing rotamer outliers.

## Weaknesses

- Why not apply the structure to a full-size ESM-Fold as a final check? The design principles are extracted entirely from a reduced setting (1-layer trunk, 1K-chain mini-dataset, single benchmark), and IPA behavior may change with depth. It's unclear if P1–P3 hold at full scale.

- MCTS + multi-agent is never compared against random sampling, greedy search, or a single-agent LLM under matched compute. Can't tell if the gains come from the searching framework design or from having an LLM generate code edits.

- P1–P3 come from post-hoc observation of one MCTS run on one reduced model, but I can not find ablation study of individual principles.

- TM-score deltas are within ±0.01 for every variant. Seems that the global fold quality is flat. NWRS improvement is driven almost entirely by lDDT, and the uniform weighting amplifies this.

---

### Official Review · Reviewer_VX4F · 2026-03-11

**Soundness:** 2
**Presentation:** 3
**Significance:** 2
**Originality:** 2
**Overall Recommendation:** 2
**Confidence:** 4

**Summary:**

The paper introduces AgentFold, a multi-agent LLM framework that automates architectural search over protein folding models. Starting from ESMFold, the system runs a propose-implement-evaluate loop organized as MCTS over a tree of code variants: an LLM planner writes architectural modifications, a debugger fixes runtime errors, and an analyst logs outcomes to a structured memory. The results show best variant improves lDDT over the baseline. The authors argue that aggregating outcomes across variants exposes design principles: soft, learnable priors injected early tend to help, while hard geometric perturbations tend to destabilize training.

**Compliance With Llm Reviewing Policy:**

Affirmed.

**Final Justification:**

Since the authors did not provide a rebuttal, my decision is unchanged.

**Key Questions For Authors:**

1. All reported results use a single-layer Folding Trunk. Do any of the discovered variants, or the design principles extracted from them, hold when the trunk depth is increased on a larger training split?

2. What fraction of LLM-proposed modifications produced runnable training jobs without intervention from the Debugger agent?

3. The paper claims MCTS improves search efficiency. How does the final set of successful variants compare to uniform random sampling over the same modification family, matched for total GPU budget?

**Limitations:**

yes

**Strengths And Weaknesses:**

**Strengths**

1. Using LLM agents to close the loop between architectural hypothesis, code implementation, and empirical evaluation is a natural direction, and protein folding provides a concrete, high-stakes testbed where the design space is genuinely large.

2. Stratifying outcomes by NWRS and tracing which intervention classes correlate with stable gains versus collapse is a reasonable approach to principle extraction from a search trajectory.

3. The paper is relatively complete in structure, with supplementary material covering evaluation protocols, metric definitions, and agent prompt templates.

**Weaknesses**

**1. The performance gains are too modest to validate the central premise.**

Even setting aside comparison with state-of-the-art systems (AlphaFold 3; IsoDDE), the paper need to frame itself as a proof-of-concept for LLM-driven architecture search on ESMFold. For that framing to be convincing, the discovered improvements should be strong enough to demonstrate that the framework surfaces something non-obvious. The observed gains are incremental and concentrated in local accuracy metrics, while global fold quality (TM-score) remains essentially unchanged. The observed improvement does not convincingly demonstrate that the search provides benefits beyond what could be achieved through simple hyperparameter tuning.

**2. The experimental setting is too simplified to support the claimed conclusions.**

All experiments use a highly reduced setting, including a single-layer Folding Trunk and limited training data. This makes it hard to judge whether the findings would transfer to more realistic models.

**3. Gains are reported without any measure of uncertainty.**

The paper reports averages, but provides no confidence intervals, standard deviations, or statistical tests. It is therefore unclear whether the improvements are meaningful or within noise.

**4. The framework does not demonstrate information gain over simpler search strategies.**

The paper claims that LLM-guided exploration improves search, but does not compare against random search, grid search, or similar human-designed modifications. As a result, it is unclear whether the method provides real guidance or just a more expensive way to constrain exploration. I agree that such information may contain useful signals for improving folding models. However, the paper does not convincingly demonstrate that its particular mechanism for injecting this information has been sufficiently designed.

---

### Official Review · Reviewer_qoJj · 2026-03-13

**Soundness:** 4
**Presentation:** 3
**Significance:** 2
**Originality:** 2
**Overall Recommendation:** 3
**Confidence:** 5

**Summary:**

In this paper, the authors introduce AgentFold, an LLM-driven multi-agent framework designed to autonomously optimize the architecture of protein structure prediction models, using ESMFold as the base architecture. The authors formulate the architectural search as a Monte Carlo Tree Search (MCTS) problem within a closed-loop, self-evolving workflow. Due to computational constraints, the experiments are conducted on a heavily reduced proxy task utilizing a 1,000-chain mini-dataset and a 1-layer ESMFold trunk. While the absolute performance of the resulting variants improves marginally over this weak baseline, the authors successfully aggregate the experimental traces to extract implicit design principles for protein folding architectures (e.g., the benefits of soft, learnable priors versus the instability of hard geometric perturbations).

**Compliance With Llm Reviewing Policy:**

Affirmed.

**Final Justification:**

I will keep my recommendation since there is no rebuttal attached.

**Key Questions For Authors:**

1. How does a generalized state-of-the-art LLM-NAS method (such as ASI-ARCH) perform on this specific ESMFold optimization task compared to AgentFold?
2. While conducting ablations on a miniature training set and a 1-layer trunk is understandable for search efficiency, can the authors provide a proof-of-concept demonstrating that their top-ranked architectural variant maintains its performance delta when scaled up to a deeper trunk (e.g., 8 or 48 blocks) and trained on the full dataset (all PDBs before 2020-05-01)?
3. Given the maximum lDDT improvement is roughly +0.05, can the authors provide variance metrics (e.g., running the baseline and the top variant across 3-5 different random seeds) to confirm that these gains are statistically significant and not an artifact of random noise?

**Limitations:**

yes

**Strengths And Weaknesses:**

Strengths:
- Protein structure prediction is one of the key tasks in biology. Automating the architectural exploration for this task is a valuable contribution to the field.
- The proposed approach is reasonable to do this task.
- The manuscript is well-structured and easy to follow.


Weaknesses:
- Lack of comparison to existing methods. The paper lacks a direct comparison to existing LLM-driven Neural Architecture Search (NAS) methods. While the authors mention ASI-ARCH in the related work, there is no empirical comparison demonstrating how AgentFold outperforms or scales differently than these existing generalized automated AI research agents on this specific task
- The improvement is very subtle. The performance gains reported are extremely subtle (e.g., the best lDDT improvement is +0.053, and NWRS improves by +0.026). Given the small scale of these deltas, it is difficult to determine whether the improvements stem from the architectural modifications themselves or merely from training noise and random initialization. Statistical significance is not established.
- The training set is too small. To manage compute costs, the authors curated a 1,000-chain mini-dataset and reduced the model to a 1-layer folding trunk. This results in a severely degraded baseline performance (e.g., an lDDT of 0.232, compared to the official ESMFold's ~0.79). It is highly questionable whether the subtle improvements—or the extracted "design principles"—discovered in this under-parameterized, data-starved toy setting will effectively transfer to a fully-scaled model.

---

### Decision · Program_Chairs · 2026-04-30

**Decision:**

Reject

**Comment:**

While reviewers agree that the paper explores an interesting direction, using multi-agent LLM systems to automate architectural exploration for protein folding, and that the presentation is clear, they raise consistent concerns about the significance and evidential strength of the results. In particular, (i) the reported improvements are very small and lack statistical validation, making it unclear whether the gains are robust; (ii) all experiments are conducted in a highly simplified setting, which limits confidence that the findings or extracted design principles transfer to realistic, full-scale models; and (iii) the proposed multi-agent and MCTS-based framework is not adequately validated against simpler baselines or existing LLM-driven search approaches.

The authors did not provide a rebuttal, so these concerns remain unaddressed. As a result, despite the paper’s promise as a proof of concept, it cannot be accepted at this time. The authors are encouraged to strengthen the empirical evidence and resubmit to a future venue.